# Generalist Vision Foundation Models for Medical Imaging: A Case Study of Segment Anything Model on Zero-Shot Medical Segmentation

**DOI:** 10.3390/diagnostics13111947

**Published:** 2023-06-02

**Authors:** Peilun Shi, Jianing Qiu, Sai Mu Dalike Abaxi, Hao Wei, Frank P.-W. Lo, Wu Yuan

**Affiliations:** 1Department of Biomedical Engineering, The Chinese University of Hong Kong, Shatin, Hong Kong SAR, China; peilunshi@cuhk.edu.hk (P.S.); jianing.qiu17@imperial.ac.uk (J.Q.); tariqabaxi@cuhk.edu.hk (S.M.D.A.); haowei@link.cuhk.edu.hk (H.W.); 2Department of Computing, Imperial College London, London SW7 2AZ, UK; 3Hamlyn Centre, Department of Surgery and Cancer, Imperial College London, London SW7 2AZ, UK; po.lo15@imperial.ac.uk

**Keywords:** Segment Anything Model (SAM), medical image segmentation, zero-shot segmentation, large AI models, foundation models, deep Learning

## Abstract

Medical image analysis plays an important role in clinical diagnosis. In this paper, we examine the recent Segment Anything Model (SAM) on medical images, and report both quantitative and qualitative zero-shot segmentation results on nine medical image segmentation benchmarks, covering various imaging modalities, such as optical coherence tomography (OCT), magnetic resonance imaging (MRI), and computed tomography (CT), as well as different applications including dermatology, ophthalmology, and radiology. Those benchmarks are representative and commonly used in model development. Our experimental results indicate that while SAM presents remarkable segmentation performance on images from the general domain, its zero-shot segmentation ability remains restricted for out-of-distribution images, e.g., medical images. In addition, SAM exhibits inconsistent zero-shot segmentation performance across different unseen medical domains. For certain structured targets, e.g., blood vessels, the zero-shot segmentation of SAM completely failed. In contrast, a simple fine-tuning of it with a small amount of data could lead to remarkable improvement of the segmentation quality, showing the great potential and feasibility of using fine-tuned SAM to achieve accurate medical image segmentation for a precision diagnostics. Our study indicates the versatility of generalist vision foundation models on medical imaging, and their great potential to achieve desired performance through fine-turning and eventually address the challenges associated with accessing large and diverse medical datasets in support of clinical diagnostics.

## 1. Introduction

Recently, large AI models (LAMs) have been actively researched as they manifest impressive performance on various downstream tasks and offer a foundation to advance and foster future research in manifold AI areas, such as computer vision and natural language processing [1,2]. In medical and healthcare domains, LAMs are also transforming methodological designs and paradigms, and establishing new state-of-the-arts and breakthroughs in various sectors including medical informatics and decision-making [3]. Despite the active development, advances in medical LAMs often lag behind their counterparts in general domains. To identify current discrepancies and guide the future development of medical LAMs, we select one of these LAMs in the general domain, i.e., Segment Anything Model (SAM) [4], which is a foundational vision model recently proposed for image segmentation and has shown stunning performance on tasks ranging from edge detection to instance segmentation, and thoroughly evaluate its zero-shot segmentation performance on medical images. Although there are few studies out there that tested SAM on medical imaging, they either only focus on one imaging modality, i.e., pathology [5], or only showcase a few qualitative segmentation samples [6] without reporting quantitative results. To provide a comprehensive and objective evaluation of SAM on medical image segmentation, this work conducted extensive experiments on nine benchmarks using the zero-shot segmentation feature of SAM. The selected datasets contain a wide diversity of medical imaging modalities and organs.

Our key findings include:SAM demonstrated better performance on endoscopic and dermoscopic images than other medical modalities, which is conjectured as SAM was trained with a large volume of RGB image data, and endoscopic and dermoscopic images are essentially images captured by RGB cameras. Therefore, when transferred to relevant medical images, SAM can demonstrate a relatively decent and consistent performance as it is tested on general RGB images.SAM failed to carry out zero-shot segmentation tasks on images that have continuous branching structures, such as blood vessels. Interestingly enough, when tested on images of tree branches, we found SAM was actually also unable to segment them in a zero-shot manner.Compared to models specially designed for medical imaging, the zero-shot segmentation capability of SAM on medical images are decent but often inferior to those domain-specific models. Our experiments reveal that the Dice coefficients of SAM on medical benchmarks were generally lower by 0.1–0.4 compared to previous state-of-the-art (SOTA) models in medical image segmentation. In the worst case, the Dice score of the zero-shot SAM is even lower than the SOTA by 0.65.Preliminary experiments of fine-tuning SAM were conducted. A simple fine-tuning of SAM on small amount of retinal vessel data led to impressive improvements of the segmentation quality, implying the great potential of SAM on medical image segmentation by fine-tuning. Our code is available by accessing https://github.com/hwei-hw/Generalist_Vision_Foundation_Models_for_Medical_Imaging.

## 2. Segment Anything Model

SAM is a generalist vision foundation model for image segmentation, and supports a diverse range of input prompts to enhance the segmentation quality. However, SAM does not recognize the type of each individual segmented object. To facilitate comparison and evaluation, the prompt for segmenting each instance is derived from the centroid of each individual ground truth mask of that instance in our study. Upon receiving the prompt, SAM generates three potential segmentation results and provides corresponding scores. The highest-scored result is selected and compared with the ground truth for evaluation. The details of implementation are introduced in Algorithm 1.   
**Algorithm 1:** SAM on Zero-Shot Medical Image Segmentation.
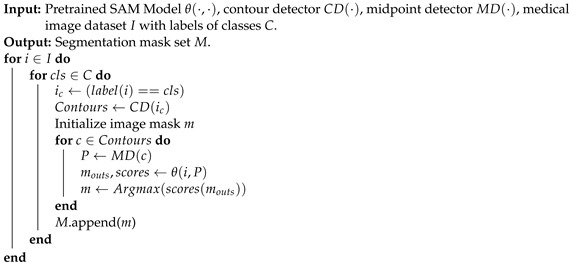


## 3. Experiments

Our study was implemented on a single NVIDIA RTX 3080 GPU and the official checkpoint of ViT-H SAM model was chosen to test the best performance of SAM on zero-shot medical image segmentation. After conducting tests across multiple medical imaging modalities, we observed that the segmentation outcomes were not consistently satisfactory among those modalities.

### 3.1. Medical Image Segmentation Datasets

Nine datasets (summarized in Table 1), including Skin Lesion Analysis Toward Melanoma Detection [7,8], Drishiti-GS [9], RIM-ONE-r3 [10], REFUGE [11], AMOS [12], MICCAI 2017 Robotic Instrument Segmentation [13], Chest X-ray [14,15], Rat Colon [16] and AROI [17] were used to examine the performance of SAM on medical image segmentation, covering a wide range of medical imaging modalities, such as OCT, MRI, and CT, as well as a diverse range of organs, including eyes, colon, spleen, kidney, gallbladder, esophagus, liver, and stomach.

### 3.2. Evaluation Metrics

To assess the zero-shot segmentation capability of SAM on medical images, two quantitative metrics were employed: Dice similarity coefficient and Intersection over Union (IoU). The Dice coefficient measures the overlap between two sets of data and ranges from 0 (no overlap) to 1 (perfect overlap). Similarly, IoU computes the ratio of the intersection over the union of two sets and ranges from 0 to 1. Both metrics were then averaged across multiple samples to obtain an overall measure of the segmentation accuracy. Specifically, the Dice coefficient and IoU were calculated as
(1)$$Dice=\frac{2|Y\cap \hat{Y}|}{\left|Y|\;+\;\right|\hat{Y}|}$$
(2)$$IoU=\frac{|Y\cap \hat{Y}|}{\left|Y|\;+\;\right|\hat{Y}|\;-\;|Y\cap \hat{Y}|}$$
where Y refers to the ground truth mask and Ŷ denotes the predicted mask of SAM based on the prompt (i.e., the centroid of the ground truth mask). Dice and IoU are both the gold standards for evaluating the overlap of the ground truth and the predicted regions in image segmentation tasks. However, IoU is more suitable for evaluating the worst-case scenarios, which could provide a more comprehensive evaluation on the medical image segmentation. As some current methods use either one of these two evaluation metrics or both to report results, we use both Dice and IoU in our work to enable direct comparison.

### 3.3. Results

This section presents the results of zero-shot medical segmentation of SAM on eight different imaging modalities, as well as preliminary results of retinal vessel segmentation by fine-tuning SAM.

#### 3.3.1. Dermoscopic Images

Skin lesion analysis dataset is sourced from the challenge hosted by the International Skin Imaging Collaboration (ISIC) [7,8]. Note that images captured by dermatoscopes with professional lighting and better magnification always present accurate details and precise contours. However, melanoma appears heterogeneously on different states of skin or diseases. Table 2 summarizes the results for comparison. From the bottom two rows of Table 2, it is evident that for the same task, there is a significant difference in the final results due to the difference in the number of tested samples. We hypothesize that this discrepancy may be attributed to the randomness from the small number of samples. Therefore, the results from testing with larger number of samples may be more reliable. SAM (1000) refers to the performance of the model assessed on the official complete testing dataset. Additionally, as shown in Table 2, the results of SAM are not competitive compared to existing methods. Specifically, SAM (259) denotes the outcome obtained by utilizing ten percent of randomly selected instances from the whole dataset as a testing subset, which facilitates a straightforward comparison with existing methods as using roughly ten percent of data has been a common practice in prior works. Figure 1A shows two samples that SAM perfectly segmented. As shown in Figure 1A, SAM is able to accurately segment the target when the anomalous regions are positioned at the center and exhibit a distinct morphology.

Figure 2A shows two failure cases. When the target lesion locates in the erythematosus or scars, the segmentation often fails due to the similarity of features between the target and the adjacent parts or the absence of clear boundaries (or in another word, the lesion is less obtrusive or apparent).

#### 3.3.2. Fundus Images

Retinal fundus images are considered as the primary modality in ophthalmic diagnosis. Specifically, optic cup (OC) and optic disc (OD) segmentation from the fundus image is the essential part. DoFE [28] is a benchmark for OC and OD segmentation that comprises three different fundus image datasets. Due to the small size of available fundus image datasets, we amalgamated the data used in DoFE to evaluate the performance of SAM. Apart from an overall evaluation, we also explored and examined the domain generalization of SAM based on the same domain partitions as DoFE.

Note that OC constitutes a whitish, cup-shaped area situated at the center of the OD, which can be approximated as two concentric circles of different radius. Specifically, the fundus images were first cropped to better observe the region of interest (ROI). The positions of two prompts exhibit a high degree of proximity, occasionally coinciding with each other. Thus, accurately segmenting between OC and OD is not easy when receiving similar prompts. This partly causes zero-shot SAM to only have half of the accuracy of current SOTAs. We hypothesize that manually prompting SAM (i.e., oracle) may lead to a better accuracy though the zero-shot mode of SAM may still lag behind SOTA by a noticeable margin.

We also investigated the performance of SAM on cross-domain scenarios following the setting in DoFE [28]. The imbalanced performance observed in Table 3 demonstrates that the domain generalization (DG) problem also exists in SAM. SAM achieves the best and worst mean performance on domain 1 and domain 4. By inspecting fundus images from four domains, we found that the degree of contrast between OC/OD and background is proportional to the final performance. However, this proportional relationship cannot be observed in DG-optimized algorithms in Table 3. This comparison reveals that SAM mainly focuses on superficial features (image contrast) instead of high-level semantics when zero-shot segmenting OC/OD in fundus images. Figure 1B and Figure 2B show some successful and failed fundus segmentation samples, respectively.

#### 3.3.3. Endoscopic OCT

For endoscopic OCT, the OCT Rat Colon dataset [16] was used, which was captured by an 800-nm ultrahigh-resolution endoscopic SD-OCT system [32]. The colonic wall is composed of three distinct layers for segmentation, namely colonic mucosa (CM), submucosa (SM), and muscularis externa (ME). A tenth of dataset was randomly sampled in our experiment to evaluate the performance of SAM. Images in this dataset do not contain much color information and complex structures, which is supposed to be easy for segmentation. Nevertheless, the zero-shot segmentation performance of SAM on endoscopic OCT is far inferior to models specifically designed for medical images as shown in Table 4. Figure 1G and Figure 2G show some segmentation samples of SAM. It can be observed that compared to successful examples of other datasets, the successful examples of endoscopic OCT are actually not perfect, while the failure cases are indeed unsatisfactory.

#### 3.3.4. Ophthalmic OCT

Annotated Retinal OCT Images Database (AROI) [34] contains 1136 annotated B-scans and associated raw high-resolution images from 24 patients with age-related macular degeneration (AMD). An ophthalmologist annotated three retinal layers and three retinal fluids in each B-scan. The official annotations on B-scan are utilized to classify eight layered structures, including the inner plexiform layer and inner nuclear layer (IPL/INL), as well as regions above internal limiting membrane (ILM) and under bruch’s membrane (BM). Pigment epithelial detachment (PED), subretinal fluid and subretinal hyperreflective material (SRF), and intraretinal fluid (IRF) are also involved for segmentation. We did not take the top layer (above ILM) into consideration (i.e., not tested), because it is not a part of internal structure of retina.

As shown in Table 5, the segmentation results for under BM are much higher than other classes, which is also confirmed by the qualitative samples shown in Figure 1H. This indicates that SAM can accurately segment the structures of clear boundaries and large continuous areas. However, the zero-shot segmentation performance of SAM is extremely unsatisfactory for small holes and elongated layered structures of irregular boundaries (such as RPE-BM), as shown in Figure 2H.

#### 3.3.5. CT

AMOS [12] contains both CT and MRI modalities for abdominal multi-organ segmentation tasks. The entire dataset introduces 500 CT and 100 MRI volume data from 15 organs in abdominal cavity. During pre-processing, we set the CT window range to [−991, 362] HU and 15,361 slices from the dataset were generated using this standard. Since the corresponding ground truth in the test set has not been released, we used the validation set which contains segmentation labels. Fifteen organs were segmented, including spleen, right kidney, left kidney, gallbladder, esophagus, liver, stomach, aorta, inferior vena cava, pancreas, right adrenal gland, left adrenal gland, duodenum, bladder, prostate and uterus. The above described are also applied to MRI and the experimental results of CT are therefore presented and discussed along with those of MRI in the next section for clarity.

#### 3.3.6. MRI

The pre-processing method of MRI data is congruent with the one used for CT data mentioned in the preceding section. A total of 3176 images originating from the validation set were obtained and selected for our case study. As the datasets of two modalities are derived from the same dataset and the 3D data were pre-processed in a same way, the average Dice scores of the two modalities are close as shown in Table 6 and Table 7, although the variations exist across individual class segmentation tasks. As shown in Figure 1C, D, SAM can accurately segment two-dimensional organs in CT and MRI images. The successful segmentation manifests the precise localization and clear demarcation, and a striking resemblance to the actual organ structures in terms of visual appearance. However, the failure cases shown in Figure 2C, D are often from the segmentation of small organs, in which SAM exhibits a tendency to segment not only the target organ itself but also the neighboring tissues.

#### 3.3.7. X-ray

The dataset (704 labelled images) used for testing SAM on X-ray is compiled from the validation and training sets of [14,15]. As shown in Figure 1F, the segmentation performance demonstrated by SAM in successful cases is impressive and the segmentation results perfectly match with the ground truth. However, the failure cases shown in Figure 2F reveal that the zero-shot chest X-ray segmentation of SAM is not always consistent. Table 8 shows the quantitative results of zero-shot X-ray image segmentation.

#### 3.3.8. Endoscopic Images

We further tested SAM on the MICCAI 2017 Robotic Instrument Segmentation dataset [35], which contains 2040 stereo camera images acquired from a da Vinci Xi surgical system on three types of segmentation task (binary, parts and instruments segmentation). The images are essentially captured by RGB cameras. As a result, the segmentation outcomes shown in Figure 1E are impressive when appropriate prompts are given. As we place the prompt at the center of each individual object, a three-class segmentation of parts is the most suitable choice for testing in order to avoid any irregular connections and interference between different parts compared with binary segmentation and instrument segmentation. The performance of SAM was tested using data from the official testing set, and the results were compared against those provided in the official report. To better visualize, segmentation results of different categories have been plotted onto the same graph for comparison, which are shown in both Figure 1E and Figure 2E. It should be noted that even though some examples exhibit visually satisfactory results, they are likely due to the predicted masks partially lying within other masks. From a quantitative perspective as shown in Table 9, SAM has already surpassed the segmentation performance of LinkNet-34 [35,36] by only receiving prompts. However, it is still not comparable to the current SOTA and there is a 30% relative gap between SAM and SOTA methods.

#### 3.3.9. Retinal Vessel in Fundus Images

We also used SAM to segment vessels in fundus images. However, the experiments revealed that zero-shot SAM is not able to accurately segment retinal blood vessels, despite the *manual* provision of additional prompts focused on areas where visibility of vessels is pronounced. At the current stage, we conjectured that the segmentation of continuously branching structures, such as blood vessels in medical images and tree branches in nature images, presents a challenge for SAM. As shown in Figure 3, it is found that SAM encounters difficulties in accurately identifying vessels as distinct segmentable objects. Therefore, we further conducted fine-tuning of SAM to examine its potential of improving segmentation results. To fine tune SAM (SAM ViT-B in this experiment), we utilized SAM adapter [37], a task-specific fine-tuning method proposed for SAM. Specifically, 20 image-mask pairs were selected from the Digital Retinal Images for Vessel Extraction (DRIVE) dataset [38] to fine-tune SAM’s mask decoder. As the fine-tuning was fully supervised, there was no prompt provided and the ground truth masks were used to supervise the training. The entire fine-tuning process was trained with the AdamW [39] optimizer. The learning rate was set to $2\times 10^{-4}$ with a total of 20 training epochs. The batch size was one with the input image size of 1024 × 1024 pixels. A total of four datasets were utilized for the quantitative assessment. Three of them are the official test sets from their respective datasets. The official training set from the STARE dataset [40] was also employed for testing as its own test set is lacking. As shown in Table 10, the fine-tuned SAM demonstrated substantial improvements in both the Dice and IoU metrics. Noticeably, the Dice scores increased by at least 200 percent relative to its zero-shot counterpart where vessels were segmented by manually providing the prompt. Furthermore, the improvements of IoU were even more pronounced. In addition, as illustrated in Figure 4. The segmentation results of SAM after fine-tuning almost perfectly match with ground truth in some cases and only some tiny parts at the terminal ends of vessels are missing, demonstrating the potential of SAM for precise medical image segmentation after a domain-specific fine-tuning.

## 4. Discussion

Through extensive experiments, it is found that SAM is unable to outperform models specially designed for medical imaging by simply using its zero-shot feature. It is worth noting that SAM is trained primarily on natural images and has limited access to diverse medical modalities containing pathological manifestations. As such, the medical segmentation results obtained using SAM’s zero-shot capability are actually decent and sometimes considered impressive as shown in Figure 1. Nevertheless, given prompts in the same way, the performance of SAM varies significantly across different medical modalities. In endoscopy and dermoscopy, SAM demonstrated better performance compared to other medical modalities. In segmenting the skin lesion, if there are no clear boundaries of the lesion or if the skin itself has pigment deposition and erythematosus, the segmentation outcome of SAM is often unsatisfactory. This phenomenon resembles the concealed patterns observed in natural images, which have been previously shown to be a challenge for SAM [43,44]. Although the retinal fundus images are also RGB images, segmenting structural targets within the internal structure of retina is not encountered in natural scenes. This has resulted in suboptimal segmentation of optic disc and complete failure in retinal vessel segmentation of SAM using zero-shot segmentation.

Despite the SA-1B dataset used for training SAM contains 1B masks and 11M images, the multiple medical domains we tested in our study are entirely unseen domains for SAM. In particular, 3D imaging predominates as a critical medical methodology, such as MRI, CT, and OCT. The 2D slices employed for analysis are the unique aspects of medical imaging and cannot be found in natural domains. For instance, the features of OCT B-scan are layered structures stretching along the entire image width, instead of closed regions. Algorithms developed specifically for the prominent features of OCT have demonstrated excellent performance [33,45]. However, SAM is unable to discriminate the tissue layers in OCT images without any prior knowledge. In addition to the presence of domain differences between medical and natural images, it has been observed in Table 3 that SAM exhibits significantly imbalanced segmentation accuracy when encountering different domain images under the same category.

To evaluate the capabilities of SAM under zero-shot settings, experiments in this study used a single prompt selection method and used the center point of the ground truth mask as the prompt for each sample. Although this approach did not fully harness the potential of SAM, it suffices to highlight the limitations of SAM in medical imaging. Recently, SAM-Adapter [37] has been proposed for tackling complex scenario segmentation on natural image datasets including but not limited to camouflaged targets and defocus blur objects, which has shown better results than SAM and other task-specific approaches. Medical images can be regarded as a distinct category of rare scenes. Consequently, it is very likely that natural image-based large models after fine-tuning may yield excellent performance on medical imaging modalities as revealed by our preliminary results of SAM fine-tuned on retinal vessel data. Meanwhile, different prompt engineering techniques can be further explored in the future. Furthermore, it is worth investigating if training a large medical vision model from scratch using only medical data can lead to a better performance than continual-training/fine-tuning a large vision model using medical images, which has been previously pretrained on a large volume of natural image data. In addition, during the development phase of large medical AI models, it is recommended to prioritize a focus on diagnostic information and invariant features present in medical images. This can potentially mitigate issues related to domain transfer, thus enhancing the overall performance and interpretability of a large AI model.

The limitations of our testing method are also worth noting, and future works are encouraged to further explore SAM on medical segmentation. Firstly, we only used one of the three prompts provided in the official setting. For some cases, the centroid might not be inside the region of interest, other segment modes of SAM should be taken into consideration, such as the bounding box mode and automatically segment everything mode, which may provide a more comprehensive evaluation of its segmentation ability. Another limitation is that we only performed fine-tuning on retinal vessel datasets. Although the results indicate that the fine-tuned model demonstrates excellent segmentation quality compared to its zero-shot counterpart, we believe experiments on a wider range of datasets are needed to examine the segmentation performance of fine-tuned SAM, which could provide a more holistic view of its pros and cons on different medical modalities.

Nevertheless, our work is the first research work that focuses on evaluating and analyzing the performance of a recently developed large AI model, i.e., SAM on a wide range of medical image segmentation tasks both qualitatively and quantitatively, with detailed comparisons between SAM and baselines. The findings from our research help identify where SAM works and how it can be fine-tuned to provide better performance on medical imaging and applications. Our results may also help guide the future development of SAM and other medical generalist AI models on domain-specific tasks. Medical image segmentation is an important and challenging task. If the advanced large models such as SAM become highly accurate and robust on medical image segmentation, either in a zero-shot fashion or after fine-tuning, they may bring in far-reaching impact and help transform and improve medical diagnostics, treatment planning, and other healthcare applications that depend on medical imaging. This could eventually lead to great benefits to both clinicians and patients. In summary, our study lays an important groundwork for developing and applying large AI models on medical image segmentation. With continued progress, such models could positively impact healthcare by assisting with and improving critical tasks in medical diagnostics. Our current work not only highlights the potential significance and societal benefits of this line of research, but also identifies the limitations and the needs of further research before the achievement of substantial real-world impacts.

## 5. Conclusions

This work presents a benchmark study of SAM on a wide range of zero-shot medical image segmentation tasks. Through comprehensive experiments, we identify the challenges that SAM currently encounters in this context. Importantly, our analysis is performed on a standardized set of prompts devoid of any prior medical knowledge, covering a diverse range of imaging modalities, such as dermoscope, fundus, CT, MRI, endoscope, X-ray, endoscopic OCT and ophthalmic OCT. The provision of precise and interactive prompts, the use of specialized feature extraction methodologies tailored for medical images, and the well-designed fine-tuning strategies of large vision models originally trained on natural images can be explored in future works. Given the unique challenges associated with medical imaging, these aspects are critical for ensuring the optimal performance of generalist models in this domain. Additionally, it is crucial that large medical vision models possess cross-domain generalizability, akin to that exhibited by physicians. This is important to avoid any negative impact on diagnostic accuracy resulting from the use of new equipment and protocols. Overall, our findings highlight the needs for medical foundation models with careful consideration given to the specific challenges posed by this complex and rapidly evolving field.

## Figures and Tables

**Figure 1 diagnostics-13-01947-f001:**
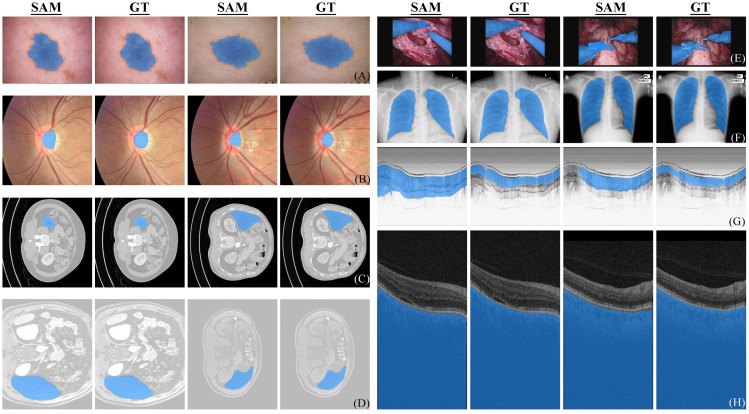
Successful segmentation examples of SAM. Eight distinct modalities labeled with (**A**–**H**) are included, corresponding to dermoscope, fundus, CT, MRI, RGB endoscope, X-ray, endoscopic OCT, and ophthalmic OCT. Each set of images comprises four images, containing two pairs of SAM segmentation versus corresponding ground truth (GT).

**Figure 2 diagnostics-13-01947-f002:**
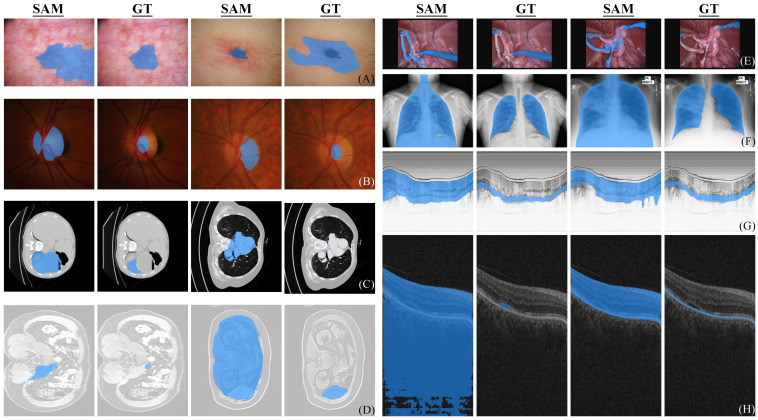
Failure segmentation examples of SAM. Eight distinct modalities labeled with (**A**–**H**) correspond to dermoscope, fundus, CT, MRI, RGB endoscope, X-ray, endoscopic OCT, and ophthalmic OCT. Each set of images consists of two paired SAM segmentation and ground truth (GT).

**Figure 3 diagnostics-13-01947-f003:**
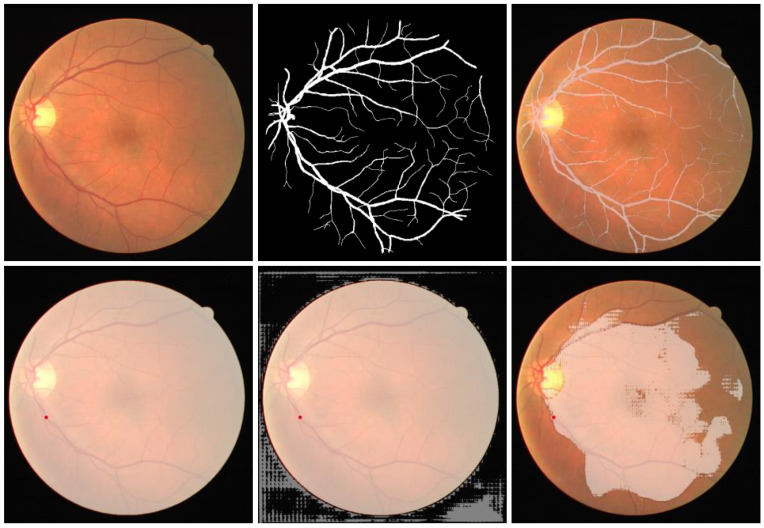
A failure sample of SAM on segmenting retinal vessels. The first row from left to right is: the initial input image, ground truth mask, and the input image superimposed with the ground truth mask. The second row from left to right shows three SAM segmented images with the score of 1.007, 0.993, and 0.673, respectively.

**Figure 4 diagnostics-13-01947-f004:**
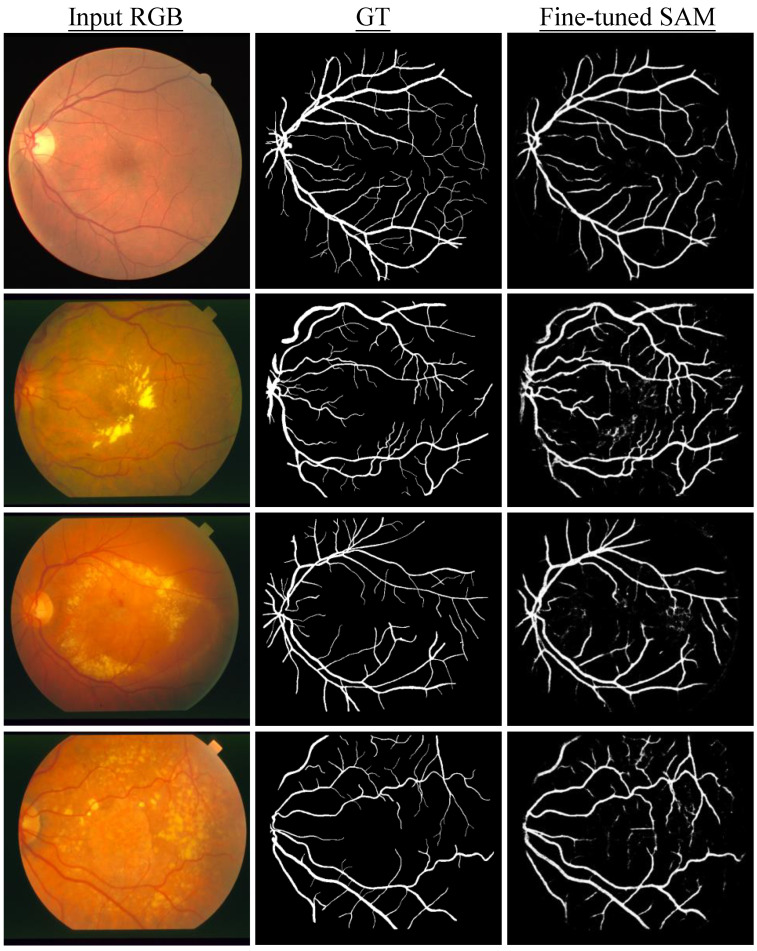
Segmentation samples of SAM fine-tuned on retinal vessels. Each row from left to right is the initial input image, ground truth mask and prediction of fine-tuned SAM. The column from top to bottom shows retinal images from four different datasets.

**Table 1 diagnostics-13-01947-t001:** Datasets Used for Examining the Zero-Shot Medical Image Segmentation Performance of SAM.

Dataset	Modality	Details	Number of Test Samples
Skin Lesion Analysis Toward Melanoma Detection [7,8]	Dermoscope	Skin	259 & 1000
Drishiti-GS [9] RIM-ONE-r3 [10] REFUGE [11]	Fundus	Eye	51 & 60 & 160
AMOS [12]	CT	Abdominal organs	15,361
AMOS [12]	MRI	Abdominal organs	3176
MICCAI 2017 Robotic Instrument Segmentation [13]	Endoscope	Tissue	1200
Chest X-ray [14,15]	X-ray	Chest	704
Rat Colon [16]	Endoscopic OCT	Colon	130
AROI [17]	SD-OCT	Retina	113

**Table 2 diagnostics-13-01947-t002:** Comparison with SOTA Methods on the Skin Lesion Analysis Toward Melanoma Detection Dataset.

Method	Dice	IoU
U-Net [18]	0.8550	0.7850
UNet++ [19]	0.8090	0.7290
CaraNet [20]	0.8700	0.782
PraNet [21]	0.8750	0.7870
TransUNet [22]	0.8800	0.8090
TransFuse [23]	0.9010	0.840
Double-UNet [24]	0.8962	0.8212
Polyp-PVT [25]	0.8962	0.8212
DuAT [26]	0.9230	0.8670
Polar Res-U-Net++(SOTA) [27]	**0.9253**	**0.8743**
SAM (259 samples, common practice)	0.6636	0.5566
SAM (1000 samples, official test set)	0.7306	0.6169

**Table 3 diagnostics-13-01947-t003:** Comparison with SOTA Methods on Fundus Datasets Evaluated using the Dice Similarity Coefficient.

Method	Domain 1		Domain 2		Domain 3		Domain 4		Mean
	OC	OD	OC	OD	OC	OD	OC	OD	Total
U-Net [18]	0.7703	0.9496	0.7821	0.8969	0.8028	0.8933	0.8474	0.9009	0.8554
Mixup [29]	0.7332	0.9297	0.7112	0.8678	0.8216	0.9042	0.8623	0.9076	0.8423
DST [30]	0.7563	0.9220	**0.8080**	0.9077	0.8432	**0.9402**	0.8624	0.9066	0.8683
JiGen [31]	0.8081	0.9503	0.7946	0.9047	0.8265	0.9194	0.8430	0.9106	0.8697
DoFE [28]	**0.8359**	**0.9559**	0.8000	**0.9837**	**0.8666**	0.9198	**0.8704**	**0.9332**	**0.8844**
SAM	0.5710	0.5563	0.5200	0.3333	0.5830	0.4157	0.3598	0.4056	0.4609

**Table 4 diagnostics-13-01947-t004:** Comparison with SOTA Methods on the OCT Rat Colon Dataset.

Method	Class	Dice	IoU
TransUNet [22]	All	**0.9265**	**-**
LiDeOCTNet [33]	All	0.9198	**-**
SAM	Colonic Mucosa	0.3491	0.2350
	Submucosa	0.2477	0.1485
	Muscularis Externa	0.2399	0.1466
	All	0.2789	0.1767

**Table 5 diagnostics-13-01947-t005:** Zero-Shot Segmentation Results of SAM for Different Classes from the AROI Dataset.

Class	Dice	IoU
ILM-IPL/INL	0.2378	0.1527
IPL/INL-RPE	0.4499	0.3153
RPE-BM	0.0688	0.0368
under BM	**0.8704**	**0.7806**
PED	0.1083	0.0673
SRF	0.1084	0.0656
IRF	0.0923	0.0548
Average	0.3237	0.2506

**Table 6 diagnostics-13-01947-t006:** Zero-Shot Segmentation Results of SAM on Different CT Organ Classes in AMOS Dataset.

Class	Dice	IoU
#1	0.1616	0.1026
#2	0.2723	0.2211
#3	**0.3465**	**0.2856**
#4	0.0943	0.0630
#5	0.1023	0.0724
#6	0.3188	0.2097
#7	0.2624	0.1812
#8	0.2946	0.2310
#9	0.1555	0.1188
#10	0.1375	0.0930
#11	0.0287	0.0202
#12	0.0585	0.0454
#13	0.1831	0.1284
#14	0.1292	0.0752
#15	0.0653	0.0360
Average	0.2105	0.1548

**Table 7 diagnostics-13-01947-t007:** Zero-Shot Segmentation Results of SAM on Different MRI Organ Classes in AMOS Dataset.

Class	Dice	IoU
#1	0.2571	0.1785
#2	**0.4622**	**0.3866**
#3	0.4434	0.3656
#4	0.1437	0.1026
#5	0.0352	0.0207
#6	0.4480	0.3311
#7	0.1475	0.0912
#8	0.2843	0.2285
#9	0.0691	0.0452
#10	0.1175	0.0754
#11	0.0054	0.0027
#12	0.0243	0.0177
#13	0.0989	0.0642
Average	0.2264	0.1723

**Table 8 diagnostics-13-01947-t008:** Quantitative Results of Zero-Shot SAM on a X-ray Dataset.

Method	Dice	IoU
SAM	0.6509	0.5136

**Table 9 diagnostics-13-01947-t009:** Comparison with SOTA Methods on the MICCAI 2017 Robotic Instrument Segmentation Dataset.

Method	Class	Dice	IoU
UNet [18]	All	0.6075	0.4841
TernausNet-16 [35]	All	**0.7597**	**0.6550**
TernausNet-11 [35]	All	0.7425	0.6223
LinkNet-34 [35]	All	0.4126	0.3455
SAM	Clasper	0.4296	0.3054
	Wrist	0.5076	0.3674
	Shaft	0.7189	0.5076
	All	0.5512	0.4227

**Table 10 diagnostics-13-01947-t010:** Quantitative Results of Fine-tuned SAM on Retinal Vessel Segmentation Datasets.

Dataset	Dice (Manual)	Dice (Fine-Tuned)	Dice Relative Increase	IoU (Manual)	IoU (Fine-Tuned)	IoU Relative Increase
DRIVE [38]	0.2267	0.7733	241%	0.1281	0.6304	392%
CHASEDB1 [41]	0.1685	0.7118	322%	0.0921	0.5526	500%
HRF [42]	0.1701	0.6776	298%	0.0931	0.5124	450%
STARE [40]	0.1902	0.7694	304%	0.1057	0.6252	491%

## Data Availability

All datasets used are publicly available, please check the respective publications.

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
