# Peer review of "Generalist Vision Foundation Models for Medical Imaging: A Case Study of Segment Anything Model on Zero-Shot Medical Segmentation"

_diagnostics, 2023, doi:10.3390/diagnostics13111947_

Round 1

Reviewer 1 Report

The authors examine the recent Segment Anything Model on medical images, and report both quantitative and qualitative zero-shot segmentation results on nine medical image segmentation benchmarks, covering various imaging modalities, such as optical coherence tomography, magnetic resonance imaging, and computed tomography, as well as different applications including dermatology, ophthalmology, and radiology. The idea is very interesting and the paper is well written. I was impressed with how fast the authors were to perform such evaluation on SAM, wich was released just a month ago. Please take into account that the following comments are intended to improve paper quality and readers' understanding.

"It should be noted that even though some examples exhibit visually satisfactory results, which may be because the predicted results of some categories overlap with each other." -> please rewrite

What was the fine tuning performed for improving vessel segmentation? Please provide more detail on that.

Did you just have to use Vit-B instead of Vit-H to have better results in this case? Please elaborate on that.

Reviewer 2 Report

The authors present the results of applying the recently published "Segment Anything Model" (SAM) to a variety of medical imaging modalities and tasks. In addition to the zero-shot performance, for one task the performance after fine-tuning is reported.

My main concern is, that the code and model presented in this article are not publicly available. It is impossible to assess the validity of the results without seeing the implementation. This is true for the zero-shot algorithm, as well as for the fine-tuning.

Another concern is that the authors use the centroid of each ground truth instance as prompt for SAM. This has multiple implications:
1. the model can not be used to make predictions on novel cases in a real zero-shot manner (where no ground truth is available)
2. for most data sets the model can not be evaluated on the test set as ground truth is not available. Therefore, scores are not directly comparable to other methods applied to the same data set (these are usually reported on the test set and their training or validation performance would also not be fair to use)
3. the centroid might not always be inside the region of interest. But (if I understand correctly), SAM will always try to segment a region containing the pixel at the prompt location.

Additional comments:
- l.112 why does this facilitate a straightforward comparison?
- l.113 how was the prompt generated for the test set?
- l.134 in extreme cases the prompt could be identical, then it would be impossible
- did you consider alternative prompt engineering techniques: e.g. using a random location from the ground truth mask (would avoid problem 3 above, but not 1 and 2).
- l.233 which prompts were used, the default ones or the manual ones mentioned in l.228?

The manuscript contains some typos.

Reviewer 3 Report

1. Discuss the significance of IoU in Evaluation Metrics section.

2. Please refer to the below article.

Zero-shot performance of the Segment Anything Model (SAM) in 2D medical imaging: A comprehensive evaluation and practical guidelines.

What is novelty in comparison to the above article?

3. How the authors performed fine-tuning?

4. Please discuss the limitation of your proposed model.

5. What is the significance of midpoint detector in your proposed algorithm.

6. How your work is useful for the society? Please discuss it in conclusion section.

Authors need to check and correct grammatical errors.

Round 2

Reviewer 3 Report

Authors have modified the article according to the review comments.